# Multi-Omic Analysis Identifies Key Genes Driving Testicular Fusion in *Spodoptera litura*

**DOI:** 10.3390/ijms26125564

**Published:** 2025-06-10

**Authors:** Yaqun Dong, Haoyun Luo, Lihua Huang, Lin Liu

**Affiliations:** Guangzhou Key Laboratory of Insect Development Regulation and Application Research, Guangdong Provincial Key Laboratory of Insect Developmental Biology and Applied Technology, Institute of Insect Science and Technology, School of Life Sciences, South China Normal University, Guangzhou 510631, China; 15625073212@163.com (Y.D.); 13070235416@163.com (H.L.)

**Keywords:** testicular fusion, testis peritoneal sheath, transcriptome, proteome, *Spodoptera litura*, *Sl3030*

## Abstract

The *Spodoptera litura*, a Lepidopteran pest known for its high fecundity, undergoes a complete metamorphosis, including a distinctive process during which the male testes fuse from two separate organs into a single entity, significantly enhancing its fertility. To elucidate the molecular mechanisms underlying this testicular fusion, this study employed an integrated multi-omics approach to investigate concurrent changes at the transcriptomic and proteomic levels. We identified a series of synchronized alterations on the peritestic larval membrane, including heme binding, peptidase activity, hydrolase activity, metal ion transport, redox reactions, and chitin metabolism, all of which are substantially enriched at specific temporal points during testicular fusion. Nine genes/proteins co-expressed at the mRNA and protein levels were selected for targeted quantitative proteomics (PRM) and quantitative PCR (qPCR) validation, leading to the identification of five genes potentially involved in the testicular fusion process: *Sl3030*, *ARCP*, *PSLRE*, *Obstructor-E*, and *Osris9B*. Notably, the gene *Sl3030*, once knocked out, not only disrupted the normal fusion process but also resulted in reduced testis size, thickened peritestic membranes, and abnormal sperm development. Transcriptomic sequencing of the *Sl3030* knockout mutant revealed its primary influence on the fusion process by affecting the assembly of the microtubule system and cytoskeleton. This research, for the first time, provides a multi-omics perspective on the response of key signaling pathways and molecular changes during the testicular fusion of *S. litura* and validates the role of the previously uncharacterized gene *Sl3030* in this process, offering valuable insights into the complex mechanisms of testicular fusion in this species.

## 1. Introduction

*Spodoptera litura* (Lepidoptera: Noctuidae) is one of the most destructive pests of agricultural crops in the Asian tropics [1,2,3]. It is polyphagous and has more than 150 host species [4]. Chemical insecticides are now the main strategy used to control the population of *Spodoptera litura* [5,6]. However, excessive pesticide usage has led to the emergence of resistance and contamination of the environment. Testicular fusion is the process during which two initially separate testes gradually come closer, adhere to each other, and ultimately fuse into a single testis during metamorphosis. A promising, effective method to control the reproduction of *Spodoptera litura* is utilizing its phenomenon of testicular fusion [7,8,9]. Studies have shown that artificially intervening in the fusion process can significantly reduce offspring sperm numbers and also decrease the egg hatch rates, thereby indicating that testicular fusion in *S. litura* can increase fertility [10]. Many Lepidoptera insects exhibit this testicular fusion phenomenon, with the exception of domesticated and wild silkworms from the family Bombycidae, though the specific timing of fusion varies slightly among species [11,12,13,14,15,16]. In *S. litura*, the testes are initially two independent and symmetrical kidney-shaped structures during the larval stage. During the prepupal stage, the two testes gradually move closer until they completely fuse into a single spherical structure [17,18,19]. This sphere further twists and rotates until the fusion boundary becomes indiscernible. The critical period for testicular fusion in *S. litura* is L6D6 15–21 h (when the two testes gradually approach each other and the peritoneal sheath starts to adhere). The robust reproductive capacity of Lepidoptera insects is closely related to the phenomenon of testicular fusion. In our past research, we found that testicular fusion is beneficial for male reproduction in *Spodoptera litura* [10,20]. Therefore, exploring the mechanism of testicular fusion of *Spodoptera litura* may help develop strategies for pest control by utilizing its reproductive characteristics.

Exploring the testicular fusion phenomenon in *S. litura* or other Lepidoptera insects is useful for understanding the mechanisms of tissue fusion in Lepidoptera and for pest control strategies.

Current research on the testicular fusion phenomenon in Lepidoptera insects focuses on several aspects. At the functional level, testicular separation surgery has demonstrated the significance of testicular fusion for individual progeny reproduction and clarified the role of testicular fusion in *Spodoptera litura*. At the bioinformatics level, the analysis of testicular fusion in *S. litura* primarily remains at the transcriptomic level, attempting to explain the key genes and pathways involved in this process at the mRNA level, such as cytoskeleton proteins, ECM/integrin interaction genes, and ECM-related proteins [7,18]. Morphologically, the process has been described by comparing the peri-testicular membrane structure between the fused testes of *S. litura* and the non-fused testes of the silkworm (*Bombyx mori*), as well as providing specific morphological descriptions of testicular fusion in other Lepidoptera insects [7,17,18].

Despite some studies on testicular fusion in Lepidoptera insects, many issues remain unresolved. Future research should further investigate the molecular mechanisms and regulatory networks of testicular fusion. Therefore, this paper employs multi-omics integrated analysis methods combined with CRISPR/Cas9 gene knockout technology to elucidate the signaling regulation of the testicular fusion process at the proteomic level more precisely. By targeting specific genes, this study aims to uncover a part of the intricate process of testicular fusion.

Proteomics technology is an invaluable tool for elucidating protein properties and profiles within complex biological systems [20,21,22,23]. This technology has been extensively employed to investigate reproduction-related proteins, particularly those involved in sexual dimorphism and gametogenesis in Lepidopteran insects such as *Bombyx mori*, *Spodoptera litura*, and *Manduca sexta* [24,25]. However, proteomic approaches have not yet been applied to study testicular fusion in many Lepidopteran species. The isobaric tags for relative and absolute quantification (iTRAQ) methodology has been widely utilized for comparative proteomic studies due to its advantages of high resolution, precise quantification, and accurate assessment of protein expression levels. Recently, parallel reaction monitoring (PRM), which offers greater precision and reliability compared to conventional mass spectrometry techniques, has been used for targeted protein analysis [26,27,28,29].

The study of testicular fusion in *S. litura* holds significant implications for advancing our understanding of insect reproductive biology, offering insights into the molecular mechanisms underlying spermatogenesis and testes development. It provides a unique perspective on the evolutionary adaptations that enhance reproductive efficiency and may reveal novel targets for pest management strategies. Moreover, elucidating the genetic and cellular processes involved in testicular fusion could contribute to broader knowledge in the fields of developmental biology and comparative physiology.

## 2. Results

### 2.1. Transcriptomic Analysis of Spodoptera litura Testicular Sheath

The relationship between the overall gene expression levels of the testis sheath at different stages of testis fusion (pre-fusion, during fusion, and post-fusion) and the fusion stage is shown in Appendix A. The violin plot indicates a broad range of gene expression distributions, with a relatively even spread across the range and a low number of highly expressed genes, resulting in a long-tail appearance. The heatmap of gene expression at different stages (Figure 1) shows distinct differences in expression levels across stages, with each stage forming a distinct cluster. Subsequent analysis will focus on those genes that are upregulated during fusion while downregulated during non-fusion stages.

To elucidate the expression patterns of genes across various developmental stages, we identified 1314 upregulated and 1112 downregulated genes during fusion compared to the pre-fusion stage (Figure 2A), and 1882 upregulated and 2092 downregulated genes during post-fusion compared to the fusion period (Figure 2B) (*p* < 0.05, |log2 fold change| > 0). Our investigation will concentrate on genes that exhibit upregulation during the fusion phase and subsequent downregulation post-fusion. GO enrichment analysis revealed that during fusion, cell component analysis mainly centered on the extracellular matrix, involving extracellular cell adhesion and chitin metabolic processes, with various hydrolases such as proteases, matrix metalloproteinases, serine proteases, and endopeptidases playing roles. Additionally, the Rho/Ras and GTPase signaling pathways were active during this period (Figure 3A). Similarly, post-fusion saw a downregulation of membrane protein and membrane component-related genes, as well as proteases, serine proteases, and endopeptidases (Figure 3B). The GO analysis suggests that during testis fusion, the testicular sheath undergoes membrane degradation, cell adhesion, and reorganization processes mediated by proteases, with the Rho/Ras and GTPase signaling pathways involved.

### 2.2. Proteomic Analysis of Spodoptera litura Testicular Sheath

Trend analysis graphs provide an intuitive display of data, revealing the different patterns of protein expression levels across various developmental stages, which aids in uncovering dynamic regulatory mechanisms. The continuous variation trends of different proteins during the pre-, during-, and post-fusion periods are categorized into eight profiles, as shown in Figure 4A. Proteins clustered in Profiles 0, 1, and 2 exhibit an increase in expression levels at the time of fusion, followed by a decrease after fusion completion; proteins in Profiles 3 and 4 show relatively consistent expression levels before and after fusion; and proteins in Profiles 5, 6, and 7 display a sustained upward trend in expression after fusion. The number of proteins in each trend category is depicted in Figure 4B, with the highest counts observed in Profiles 1, 2, 5, and 6. Specifically, Profile 1 clusters 132 proteins with a *p*-value of 4.6 × 10^−24^, indicating a significant difference. Given the characteristic expression pattern of Profile 1 proteins, which are upregulated exclusively during testis fusion, it is hypothesized that these proteins play a more critical role in the fusion process. The number of differentially expressed proteins during the occurrence of testis fusion (Figure 4C) is notably higher than the number after fusion (Figure 4D), suggesting that a greater number of differential proteins are involved in the fusion process. This also implies that some proteins synthesized for testis fusion may continue to function after the process has taken place.

GO enrichment analysis of significantly different proteins before and during fusion revealed that pre-fusion proteins clustered mainly in chitin binding and metabolism, nutrient metabolism, hydrolase activity (e.g., endopeptidase, serine protease, metalloproteinase), signal transduction, and cytoskeletal protein binding (Appendix A). In contrast, post-fusion proteins were primarily involved in initial upstream processes like nucleic acid binding and protein folding, suggesting these proteins may participate in early regulatory events of fusion and with a corresponding decrease in expression once fusion is complete (Appendix A). Nonetheless, we identified certain protein categories that exhibited an upregulation prior to fusion and a downregulation during post-fusion, including nutrient metabolism, protein metabolism, hydrolase activity, serine protease activity, chitin binding and metabolism, actin binding, and nucleic acid binding proteins (Figure 5A). Notably, the number of differentially expressed proteins related to hydrolysis and the extracellular matrix was similar before and after the fusion period, whereas proteins associated with chitin metabolism, nutrient metabolism, transcription factors, and ATP binding predominantly showed higher quantities during fusion/pre-fusion compared post-fusion/fusion, indicating greater protein level fluctuations prior to fusion, consistent with the results depicted in Figure 4B,C. Specific proteins involved in these categories are listed in Figure 5B. Proteins significantly upregulated during fusion were enriched in pathways related to apoptosis, lysosome, peroxisome, cell junction (e.g., adhesive junction, intercellular junction, tight junction), and cell signal transduction pathways (e.g., AMPK, FoxO, HIF1, Hedgehog, Hippo, MAPK) (Figure 5C), with key proteins illustrated in Figure 5D.

### 2.3. Combined Transcriptomic and Proteomic Analysis of the Testis Sheath

Using a nine-quadrant diagram, we represented the relationship between mRNA and protein levels in the testicular sheath before and after testis fusion. Quadrants III and VII indicate a consistent trend in mRNA and protein expression, showing upregulation or downregulation. Quadrants I and IX suggest a negative correlation between protein and mRNA levels, indicating post-transcriptional or translational regulation. Quadrants II and VIII show differential mRNA expression without corresponding changes in protein levels, suggesting post-transcriptional or translational regulation. Quadrants IV and VI indicate differential protein expression without changes in mRNA levels, suggesting translational regulation or protein accumulation. As shown in Figure 6A, the Pearson correlation coefficient between mRNA and protein levels before and after testis fusion is 0.5995 (*p* = 7.731 × 10^−63^), indicating moderate correlation. We focus on the synchronous changes in mRNA and protein in Quadrants III and VII. Similarly, the co-expression of mRNA and protein after testis fusion is shown in Figure 6B, with a Pearson correlation coefficient of 0.3029 (*p* = 2.736 × 10^−18^), indicating a weak correlation.

At the transcriptomic level, 761 genes were upregulated and 548 downregulated during testis fusion compared to pre-fusion (*Sl* Fusion vs. Pre-fusion). After fusion, 620 genes were upregulated and 970 downregulated compared to during fusion (*Sl* Post-fusion vs. Fusion). At the proteomic level, 117 proteins were upregulated and 134 downregulated during testis fusion compared to pre-fusion (*Sl* Fusion vs. Pre-fusion). After fusion, 289 proteins were upregulated and 201 downregulated compared to during fusion (*Sl* Post-fusion vs. Fusion) (Figure 6C). Cross-analysis identified 72 co-expressed genes and proteins before fusion (*Sl* Fusion vs. Pre-fusion) (Figure 6D) and 71 co-expressed genes and proteins after fusion (*Sl* Post-fusion vs. Fusion) (Figure 6E), showing differential expression at both mRNA and protein levels, thus warranting detailed analysis.

GO enrichment analysis of the 72 co-expressed genes and proteins before fusion revealed significant enrichment in molecular functions such as catalytic activity, oxidoreductase activity, chitin binding, ATPase activity, and hydrolase activity. In cellular components, they were primarily associated with extracellular regions and membrane components. Biological processes included chitin metabolism, redox reactions, serine family metabolism, and other metabolic processes (Appendix A). Similarly, GO enrichment analysis of the 71 co-expressed genes and proteins after fusion showed differences in molecular functions such as heme binding, oxidoreductase activity, chitin binding, ATPase activity, and peptidase activity. Cellular components included extracellular regions, membrane components, and the cytoskeleton. Biological processes involved chitin metabolism, redox reactions, glucosamine metabolism, cGMP metabolism, and other metabolic processes (Appendix A). Comparing the differential genes in both processes revealed consistent major gene classifications such as heme binding, peptidase activity, hydrolase activity, metal ion binding and transport, redox reactions, and chitin metabolism, with similar numbers of enriched genes, indicating significant differential expression of these key genes before and after fusion to facilitate testis fusion.

Parallel reaction monitoring (PRM) is a mass spectrometry-based targeted quantitative technique that allows the quantification of proteins without antibodies and post-translational modifications. PRM offers high sensitivity, specificity, and a dynamic range of over four orders of magnitude, enabling precise protein quantification. It distinguishes interference signals from true signals, providing higher selectivity for target proteins. To investigate genes and proteins involved in testis fusion, 72 co-expressed genes and proteins before fusion (cor-DEGs-DEPs Fusion vs. Pre-fusion) and 71 after fusion (cor-DEGs-DEPs Post-fusion vs. Fusion) were cross-analyzed (Figure 7A), identifying 29 common genes and proteins significantly differentially expressed before and after fusion (Figure 7B). PRM validation of these 29 proteins revealed that 14 proteins, including *Sl*3030, Dusky, Pollen-specific leucine-rich repeat extension-like protein 1, protein splits ends-like, protein obstructor-E, zonadhesin, carboxypeptidase B-like, and uncharacterized LOC111357321, LOC111356160, LOC111353918, unconventional myosin-XV, mucin-3A, protein obstructor-E-like, showed consistent quantification results across RNA-seq, iTRAQ, and PRM, indicating high expression during testis fusion and low expression before and after fusion (Figure 7C). Information on these 14 proteins is presented in Table 1. PRM validation confirmed that 13 *Spodoptera litura* proteins exhibited the characteristic of high expression only during testis fusion (prepupal stage). To validate gene expression levels, the top 9 genes with the highest expression changes were selected for real-time quantitative PCR to analyze their temporal expression patterns, tissue-specific expression, testis localization, and expression patterns during key fusion periods and compared with homologous genes in *Bombyx mori*.

Testis sheath samples were collected before fusion (L6D3), during fusion (L6D6), and after fusion (P3), and RNA was extracted and reverse transcribed to cDNA. Real-time quantitative PCR validation showed that *Sl3030*, *ARCP*, *PSLRE*, *Dusky*, *Obstructor-E*, *Osris9B*, and *ANPCE* exhibited high expression during testis fusion and low expression before and after fusion (Figure 8). These seven genes were further analyzed for testis tissue-specific expression. Testes were collected at L6D6, and testis sheath and spermatocyte were isolated. RNA was extracted from both testis sheath and spermatocyte, reverse transcribed to cDNA, and purity was verified using *Slcollagen* (testis sheath-specific) and Slvasa (spermatocyte-specific) genes to ensure no cross-contamination. Real-time quantitative PCR showed that five genes, *Sl3030*, *ARCP, PSLRE*, *Obstructor-E*, and *Osris9B*, were specifically expressed in the testis sheath and had low expression in spermatocytes, suggesting their involvement in testis sheath function during fusion (Figure 8B).

Considering *Bombyx mori* as a non-fusion testis lepidopteran, we examined the expression patterns of homologous genes *Sl3030*, *ARCP*, *PSLRE*, *Obstructor-E*, and *Osris9B*. Results showed that these genes did not exhibit high expression only during the pupal stage in *Bombyx mori*, unlike in *Spodoptera litura*, suggesting their potential role in testis fusion (Appendix A). Testis fusion timing varies among lepidopteran species, with *Spodoptera litura* testis fusion occurring on the last day of the last instar larva (L6D6) around 15–21 h. Testicular sheath samples were collected at L6D6 0 h, 6 h, 12 h, 15 h, 18 h, 21 h, 24 h, and PD1–3 h, 6 h for Q-PCR validation. Expression patterns fell into two categories: the first showed specific high expression only during the critical testis fusion period (15–24 h) and no expression before 15 h or after 24 h, such as *Sl3030* and *Osris9B*; the second showed a gradient increase in expression from L6D6 0 h, peaking at the critical fusion point (L6D6 18 h) and ceasing expression post-fusion. These patterns indicate gene function differences but share a peak expression at the critical fusion point (18 h) and cessation post-fusion (Figure 9).

### 2.4. Phenotypic Analysis of CRISPR/Cas 9 Gene Knockout

The *S. litura* gene *Sl3030*, identified from both transcriptome and proteome analyses of the testicular sheath during the fusion process, exhibits high expression during the testicular fusion. Quantitative PCR results reveal that this gene is specifically highly expressed in the peritoneal sheath and shows a gradient increase in expression within 24 h of the prepupal stage, peaking at the time of testicular fusion (L6D6-18 h). The gene’s expression is period-specific (highly expressed during testicular fusion) and testis-localized (highly expressed in the peritoneal sheath). In contrast, its ortholog in the silkworm shows a different expression pattern and is not highly expressed in the peritoneal sheath. This suggests that *Sl3030* may be associated with the testicular fusion process in *S. litura*. To explore the function of this gene, sgRNA for *Sl3030* was designed, and CRISPR/Cas9 gene knockout was performed to observe the effects on testicular fusion and reproduction after gene mutation.

The *Sl*3030 protein consists of 316 amino acids, includes three disordered regions, and has a transmembrane domain at the terminus (predicted by SMART, Appendix A). Intrinsically disordered proteins (IDPs) lack a defined three-dimensional structure under natural conditions but possess normal biological functions and are involved in crucial physiological and pathological processes such as signal transduction, DNA transcription, cell division, and protein aggregation. IDPs are characterized by low sequence complexity. The cellular localization prediction suggests that *Sl*3030 protein is localized within the nucleus (predicted by PredictProtein; Cell-PLoc 2.0), with DNA-binding sites at amino acid positions 20–46, 60–74, 126–169, 295–299, and an RNA-binding site at positions 37–41 (Appendix A). It is speculated that this protein may act as a transcription factor with regulatory functions within the nucleus. The homologous protein of *Sl*3030 in the cotton bollworm is annotated as forminA, with a sequence similarity of 83.65%. However, since this protein does not contain the conserved FH2 domain of the formin family, it is not a member of the formin family and is therefore named *Sl*3030 (*S. litura* uncharacterized LOC111363030). Protein sequence alignment of the uncharacterized protein *Sl*3030 with model organisms such as *Mus musculus*, *Xenopus laevis*, *Danio rerio*, *Gallus gallus*, and *Caenorhabditis elegans* indicates a possible involvement in the transcriptional regulation of RNA (Appendix A).

To investigate the function of the *Sl3030* gene, CRISPR/Cas9-mediated gene knockout of *Sl3030* was performed to observe the impact of gene deletion on testicular fusion and reproduction. The entire gene sequence, CDS sequence, and mRNA sequence of *Sl3030* were searched on NCBI, with the gene identified as having three exons and two introns. Three sgRNAs were designed within the first exon, and in vitro CRISPR/Cas9 assays demonstrated that the second sgRNA site effectively cleaved the target sequence. Consequently, the sgRNA sequence from the second exon was selected for further application. Following the CRISPR/Cas9 gene knockout, bi-allelic mutations were successfully detected near the target site in the mutants. The G2 generation preserved a gene mutation characterized by a 63 bp deletion, denoted as *Sl3030*^−63bp/−63bp^ (Appendix A). Compared to the wild type, the mutant with this deletion exhibited a shortened second disordered region by 21 amino acids, leading to alterations in its three-dimensional structure (Appendix A).

In wild-type (WT) and *Sl3030*^−/−^ homozygous mutant adult *Spodoptera litura* (AD3), the testes of mutants were smaller in volume compared to the wild-type (Appendix A). Both the long and short axes of the testes in homozygous mutants were shorter than those in the wild-type (Appendix A). Histological sections revealed that the seminiferous tubule membranes in the mutant testes were disorganized, with tightly packed spermatocyst clusters showing higher cell density and no clear separation between cells, making it difficult to distinguish mature sperm bundles (Appendix A).

In three-day-old adult homozygous mutants (AD3), the testes appeared partially unfused. While the two testes were adjacent, they did not twist and fuse into a single spherical structure. A significant fissure was observed in the middle of the testicular complex, with an inward indentation visible at the base of the complex (Figure 10A–F). This phenotype was observed in 40% of cases. Histological examination of the abnormal testes showed that the internal spermatocytes were abnormally developed. In contrast to the uniform distribution of mature sperm bundles in wild-type testes, with clear gaps between bundles, mutant testes had tightly packed spermatocytes. High-magnification imaging revealed that these spermatocytes were still in the early spermatocyst stage, not having formed mature sperm bundles (Figure 10G).

Compared to the wild-type, the peritoneal membranes of *Sl*3030 mutant testes also exhibited significant changes. The thickness of the fusion surface of the peritoneal membrane in the mutant testes was notably increased compared to the wild-type (Figure 10H,I). During normal fusion, the peritoneal membrane thins as the outer membrane degrades, facilitating the fusion process. However, in mutants, the reduction in the thickness of the fusion surface was not observed, impeding the fusion process to some extent.

### 2.5. Transcriptomic Analysis of Spodoptera litura Testis Sheath: Mechanistic Investigation of Sl3030 Mutant Testis Fusion

To explore the mechanistic impact of *Sl3030* deletion mutations on testis fusion, transcriptomic sequencing was performed on the prepupal testis of both wild-type and mutant *Spodoptera litura*. A total of 608 co-expressed genes were identified between the wild-type and mutant. Comparative analysis revealed 384 upregulated and 224 downregulated genes in the wild-type compared to the mutant (Figure 11A,B).

GO enrichment analysis of the downregulated genes due to gene deletion indicated involvement in cilia, cytoskeleton, and microtubule systems, including dynein assembly, activity, and migration. This suggests that the *Sl3030* gene facilitates testis fusion by influencing cytoskeletal assembly and activity. The knockout of this gene results in diminished activity of the testicular sheath (Figure 11C).

Additionally, KEGG pathway analysis identified the Hedgehog signaling pathway, which further supports this finding (Appendix A). The Hedgehog signaling pathway plays a crucial role in regulating cell motility. In vertebrates, cilia serve as essential centers for Hedgehog signal transduction, and the Hedgehog signaling process is closely related to ciliary movement. Cilia are cell structures composed of microtubules, with tubulin being the fundamental protein constituting microtubules. This signaling pathway can influence cell migration and tissue formation.

## 3. Discussion

From a transcriptomic perspective, the testicular sheath of *S. litura* is subjected to a series of morphological changes during the process of testis fusion, including membrane degradation, enhanced cell adhesion, and tissue reorganization. These events are orchestrated by the action of proteolytic enzymes and are under the regulatory influence of the Rho/Ras and GTPase signaling cascades, which are pivotal in modulating these dynamic cellular processes. Prior research indeed confirms that during the testicular fusion process in *Spodoptera litura* [7,18], morphological changes such as membrane degradation, enhanced cell adhesion, and tissue reorganization occur, aligning with the perspective presented in this paper. Matrix metalloproteinases (MMPs) in extracellular matrix (ECM) degradation is consistent with the orchestrated events mentioned in the proposed viewpoint, involving proteolytic enzymes, including MMP [30]. However, the regulatory influence of the Rho/Ras and GTPase signaling pathways in this process is a unique aspect of the viewpoint presented here.

From a proteomic perspective, specific protein groups showed an upregulation before testicular fusion and subsequent downregulation post-fusion, encompassing key processes such as nutrient and protein metabolism, hydrolytic activities including serine proteases, chitin binding and metabolism, actin binding, and nucleic acid binding. Integrating multi-omics analyses, we scrutinized the transcriptomic and proteomic changes during the testicular fusion process in *Spodoptera litura*. Our focus was on genes and signaling pathways that exhibited consistently high expression at both the transcriptomic and proteomic levels. Subsequent validation through protein mass spectrometry and quantitative PCR confirmed the crucial roles of *Sl3030* in testicular fusion and offspring reproduction in *S. litura*. *Sl3030*, a protein with an unknown function, was revealed to influence normal testicular fusion, resulting in reduced testis size, thickened peri-testicular membranes, and abnormal sperm development. This gene not only regulates the fusion process of the testes but also significantly impacts the reproductive capacity of the offspring.

Our exploration into the reasons behind the non-fusion phenotype observed in *Sl3030*-deficient mutants suggests that the absence of *Sl3030* significantly impacts the assembly of the cytoskeleton and the activity of genes associated with the microtubule system, thereby affecting the normal process of membrane fusion. Although numerous previous studies have confirmed that 20E plays a regulatory role in testicular fusion processes, the *Sl3030* gene is not regulated by 20E in vitro, indicating that *Sl3030* does not participate in testicular fusion through the direct 20E signaling pathway. Sequence analysis suggests that *Sl3030* is a disordered protein with multiple low-complexity regions located in the nucleus, implying that this protein may undergo flexible structural changes to rapidly respond to intracellular signals. Analysis of the transcriptome from mutants revealed that genes related to the assembly of the cytoskeleton and the activity of the microtubule system were significantly affected after the loss of *Sl3030*, suggesting that *Sl3030* may regulate the microtubule system, influencing the extension of actin and cell migration, thereby preventing membrane fusion.

In *Sl3030* knockout homozygous mutants, genes associated with cilia, cytoskeleton, and microtubule system, including the assembly, activity, and migration of kinesins, were significantly downregulated, thus affecting the normal process of testicular fusion. The results demonstrate the indispensable role of the microtubule system in the process of testicular fusion, indicating the direction in which *Sl3030* affects testicular fusion and laying a theoretical foundation for subsequent studies on how *Sl3030* affects testicular fusion through participation in the microtubule system. The microtubule system is a key component of the cytoskeleton and plays a fundamental role in various cellular processes, such as cell migration and tissue remodeling. Microtubules are polar structures composed of α- and β-tubulin heterodimers, which undergo continuous processes of polymerization and depolymerization, known as dynamic instability. This dynamic behavior is crucial for cell migration, and in insect cells, microtubule-associated proteins (MAPs) such as Tau and Doublecortin regulate cell polarity and directed movement by controlling the microtubule system, which is essential for cell migration. The interaction between microtubules and the cell membrane is mediated by cell adhesion, where integrins and other receptors convert extracellular matrix (ECM) signals into intracellular responses.

The microtubule (MT) system, integral to the cytoskeleton, is essential for a variety of cellular processes, including cell migration and tissue remodeling [31,32,33]. Composed of α- and β-tubulin heterodimers, microtubules exhibit dynamic instability, characterized by continuous polymerization and depolymerization. This dynamic nature is vital for cell migration, with the coordinated assembly and disassembly of microtubules at the cell’s leading edge facilitating directed movement. In insect cells, microtubule-associated proteins (MAPs), such as Tau and Doublecortin, modulate this instability, influencing cell polarity and migration efficiency by regulating microtubule organization [34,35,36,37].

Cytoskeleton-related genes encode proteins that interact with microtubules to manage cellular responses during tissue restructuring. Insect-specific genes, including *spindle* (*spn*) and *non-claret disjunctional* (*ncd*), are critical in the reorganization of microtubule networks throughout development [38,39,40,41]. The proteins encoded by these genes promote microtubule dynamics and anchoring, which are essential for cellular repositioning and morphogenesis. Additionally, the interaction between microtubules and other cytoskeletal elements, such as actin filaments and intermediate filaments, is facilitated by crosslinking proteins like spectraplakins, which maintain structural integrity and synchronize cellular remodeling [42]. The connection between microtubules and the cell membrane, mediated through focal adhesions, allows extracellular matrix (ECM) signals to be transduced into intracellular responses via integrins and other receptors [43,44,45]. In insects, Rho family GTPases, including RhoA, Rac1, and Cdc42, are instrumental in regulating cytoskeletal dynamics [46,47,48]. Their activation is crucial for the orchestration of microtubule and actin cytoskeleton reorganization, which underpins processes such as wound healing and morphogenesis [49,50].

The exploration of testicular fusion in the lepidopteran insect *Spodoptera litura* has unveiled pivotal molecular underpinnings of this physiological phenomenon. The studies by Chen et al. in 2020 and 2023 [7,17] have, respectively, contributed to our understanding of the intricate mechanisms at play during the larva-to-pupa transformation, where the coalescence of testes is deemed essential for spermatogenesis and reproductive success.

The 2023 study by Chen et al. [17], through meticulous identification and expression analysis of integrin subunits, shed light on the role of these cell adhesion receptors in the testicular fusion process. The collective findings from these studies underscore the complexity of testicular fusion, highlighting the interplay between the cytoskeleton, ECM, and cell adhesion molecules. These revelations not only substantiate the hypothesis regarding the role of the microtubule system in cellular processes but also pave the way for future research aimed at unraveling the molecular nuances of tissue morphogenesis and organ development in insects.

We can provide a scaffold for further investigation into how cytoskeletal dynamics interface with microtubule functions during testicular fusion, although we do not directly address the microtubule system’s involvement. The identification of DEGs and their association with cell migration, adhesion, and fusion processes offers a foundation to explore the broader implications of cytoskeletal remodeling in developmental biology and may reveal novel targets for pest management strategies in agriculture in future research.

To summarize, we identified signaling pathways involved with testicular fusion in *S. litura* and the concomitant changes in genes/proteins at both the RNA and protein levels, with a particular focus on validating the impact of the *Sl3030* gene. This outcome holds significant theoretical and practical implications for comprehending the mechanisms underlying testicular fusion in the *Spodoptera litura*, offering valuable insights for both basic research and potential applications in pest management strategies.

## 4. Materials and Methods

### 4.1. Spodoptera litura Feeding and Sample Collection for Transcriptome Analysis

The experimental insects, *Spodoptera litura,* were cultivated in a controlled environment at 26–28 °C, with a humidity of 60–70%, under a 12-h light/dark cycle. Adult moths were nourished with a 10% honey solution within an artificial breeding chamber. Testes were harvested from *S. litura* at distinct developmental phases—the 6th instar day 3 (L6D3), the prepupal (PP), and the pupal day 3 (P3), to isolate the peritoneal sheath. Following the removal of the testes from the abdominal cavity, they were promptly placed into an Eppendorf tube containing 100 µL of phosphate buffer, and spermatocytes were extracted by gently agitating with a grinding rod. The testicular peritoneal sheath was meticulously excised using forceps and rinsed thrice with phosphate-buffered saline (PBS) to eliminate any remaining spermatocytes. This procedure was conducted in triplicate for each developmental stage.

### 4.2. Spodoptera litura Sample Collection for Protomic Analysis

On the 6th instar day 6 (L6D6), the testes of *S. litura* were collected at three time points: 15–18 h (the right and left symmetrical testes merely touching); 18–21 h (the contact surfaces of the two testes are fused into a straight line, and the upper and lower ends of the testis are not fused); 21–24 h (the two testes completely fuse into a spherical shape, contact surfaces disappear). After separating the peritoneal sheath from the sperm cells, the peritoneal sheath was rapidly frozen in liquid N_2_ and kept at 80 °C for iTRAQ analysis and PRM testing. The prepared peritoneal sheath was ground into powder in liquid nitrogen, and the powder was dissolved in 1 mL lysis buffer 4% *w*/*v* SDS, 1 mM DTT, 150 mM Tris-HCl, pH 8. After being sonicated, the homogenate was centrifuged at 12,000× *g* for 20 min at 4 °C to separate the supernatant. The supernatant was divided into 4 EP tubes (1.5 mL); each tube was about 250 μL, and 1 mL of acetone was added to precipitate overnight at −20 °C followed by centrifugation as above. The extract was dissolved in 200 µL of lysis buffer, and the protein concentration was measured by BCA assay.

### 4.3. Protein Digestion and iTRAQ Labeling

A 200 μg total protein sample was processed using the methanol/chloroform protocol for reduction, alkylation, and precipitation. Each sample was first mixed with 4 μL of reducing reagent and incubated at 60 °C for 1 h. Then, 2 μL of cysteine-blocking solution was added, and the sample was left at room temperature for 30 min. The protein solution was transferred to a 10 KD ultrafiltration tube (Merck, Darmstadt, Germany), which was then centrifuged at 12,000× *g* for 15 min. The solution in the collection tube was discarded. Next, 100 μL of 8 M urea (pH 8.5) was added, followed by centrifugation at 12,000× *g* for 20 min at 4 °C. This washing step was repeated twice. After removing the solution again, 100 μL of 0.25 M TEAB (pH 8.5) was added. The collection tube was replaced with a fresh one, and the ultrafiltration tube was filled with 50 μL of 0.5 M TEAB. Trypsin was added to the mixture.

The following day, additional trypsin was added to achieve a trypsin-to-protein ratio of 1:100. The mixture was incubated at 37 °C for 4 h. After digestion, the ultrafiltration tube was centrifuged at 12,000× *g* for 20 min. The digested peptide solution was collected from the bottom of the collection tube. Then, 50 μL of 0.5 M TEAB was added to the ultrafiltration tube and centrifuged at 12,000× *g* for 20 min at 4 °C. Combining this with the previous step yielded 100 μL of enzymatic hydrolyzed sample.

For labeling, the samples were designated as follows: (before fusion1)-114, (before fusion2)-115, (on fusing1)-116, (on fusing2)-117, (after fusion1)-118, and (after fusion2)-119.

### 4.4. Designing the Target Site and Crafting the Single Guide RNA (sgRNA) for Injection Purposes

Following the 5′-GG-(N)18-NGG-3′ target sequence rule, the exon 1 sgRNA target site of *Sl3030* was pinpointed. Utilizing a PCR-based technique, sgRNA was generated by a previously detailed protocol [8]. In summary, a pair of primers was crafted to amplify an sgRNA fragment measuring 114 bp; the forward primer, sgRNA-F, included a T7 polymerase-binding site along with the sgRNA target sequence, while the reverse primer, sgRNA-R, encompassed the rest of the sgRNA sequences. These primers were melted and annealed via PCR to create a template DNA. The PCR system consisted of 25 μL of 2× PrimerSTAR Max Premix (TaKaRa, Dalian, China), 2.5 μL each of the forward primer sgRNA-F and the reverse primer sgRNA-R, complemented by 20 μL of ddH_2_O. The PCR conditions were as follows: an initial denaturation at 98 °C for 5 min, then 32 cycles of 98 °C for 10 s, 55 °C for 150 s, and 72 °C for 15 s, concluding with a final extension at 72 °C for 10 min. The resultant purified PCR product served as the template for in vitro transcription, which was carried out using the MEGAScript T7 kit (Ambion, Austin, TX, USA) following the manufacturer’s guidelines.

### 4.5. Injection of Cas9/sgRNA Complexes and Subsequent Mutation Analysis

A volume of 9.6 nL, in which sgRNA and Cas9 protein were thoroughly combined to achieve a concentration of 300 ng/μL each, was microinjected into eggs laid within the previous 2 h using an ultra-fine injector (FemtoJet, Germany, Hamburg). Post-injection, the eggs were maintained at a temperature of 25 ± 1 °C for a period of three days until they hatched.

To ascertain the occurrence of mutations at the designated target sites in the initial generation (G0), a subset of the eggs was selected one-day post-injection for genomic DNA (gDNA) extraction, followed by PCR amplification using gene-specific primers (refer to Appendix A) and subsequent sequencing. As the G0 generation of *S. litura* reached the pupal stage, the shed skins from the 6th instar larvae were gathered at random to extract gDNA for PCR amplification and sequencing.

Individuals exhibiting indel mutations were then mated with wild-type (WT) moths to produce the first filial generation (G1). From each G1 offspring group, roughly 30 eggs were harvested and pooled for genomic DNA extraction to screen for mutations via PCR. The amplified PCR products were purified, cloned into the pMD18-T vector (Takara, Japan), and sequenced to pinpoint the precise nature of the indels. During the pupal phase, a random selection of 30 individuals from each group was subjected to DNA mutation screening from their shed skins. Moths with identical mutant genotypes were mated to generate the second filial generation (G2). A homozygous G2 population characterized by a 63 bp deletion in the *Sl3030* gene was chosen for further propagation. The impact of these mutations on the protein was assessed using MEGA 7.0 Software by aligning the codons of the WT and mutant sequences [51].

### 4.6. Data Analysis Methods

#### 4.6.1. Fertility Statistics of CRISPR/Cas9 Gene Knockout Mutants

Mating was performed between wild-type males and females, as well as between wild-type males and mutant females, wild-type females and mutant males, and mutant males and females (n = 5 × 5 for each cross). To assess fecundity, eggs laid within the first three days post-mating were counted. Since the *S. litura* moths produce multi-layered egg clusters, a soft brush was used to gently disperse the clusters, and the eggs were then counted under a microscope.

#### 4.6.2. Measurement of Testicular Size and Thickness of the Peritoneal Sheath

The length and width of the testes, as well as the thickness of the peritoneal sheath in tissue sections, were measured using Image-Pro Plus 6.0. Each sample was measured in triplicate, and the data were analyzed for statistical significance using GraphPad Prism 8.

## Figures and Tables

**Figure 1 ijms-26-05564-f001:**
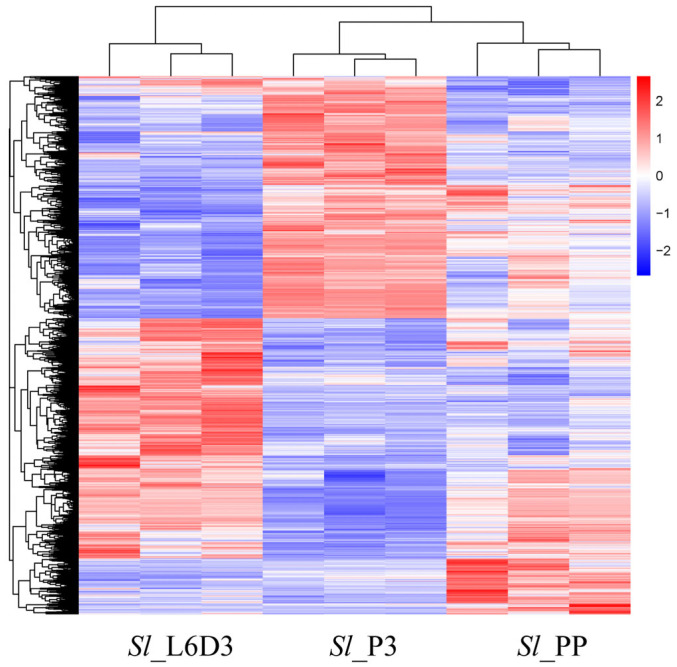
Basic RNA-Seq data of the *S. litura* peritoneal sheath. Heat diagram representing the differentially expressed genes (DEGs) at different stages of *S. litura*.

**Figure 2 ijms-26-05564-f002:**
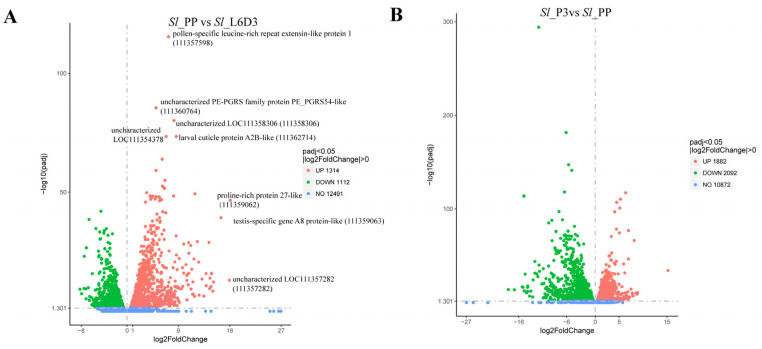
(**A**) Volcano plot illustrating differential gene expression during testicular fusion stage versus pre-fusion stage. Red dots indicate genes up-regulated during testis fusion, and green dots indicate genes down-regulated during testis fusion. (**B**) Volcano plot of differential gene expression during post-fusion stage versus fusion stage. Red dots indicate genes up-regulated after testicular fusion, and green dots indicate genes down-regulated after testicular fusion.

**Figure 3 ijms-26-05564-f003:**
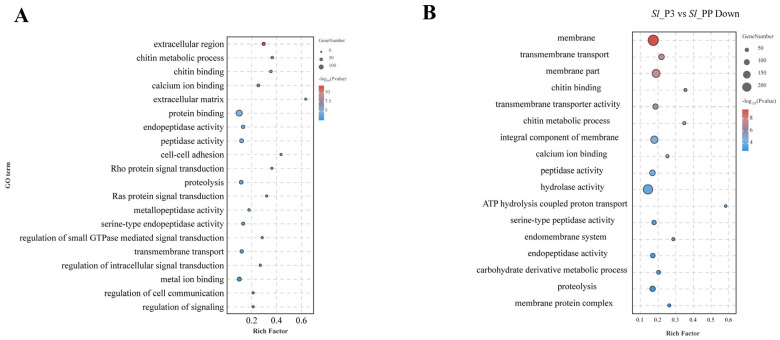
(**A**) GO enrichment of upregulated genes during fusion vs. pre-fusion period. (**B**) GO enrichment of downregulated genes during post-fusion vs. fusion period.

**Figure 4 ijms-26-05564-f004:**
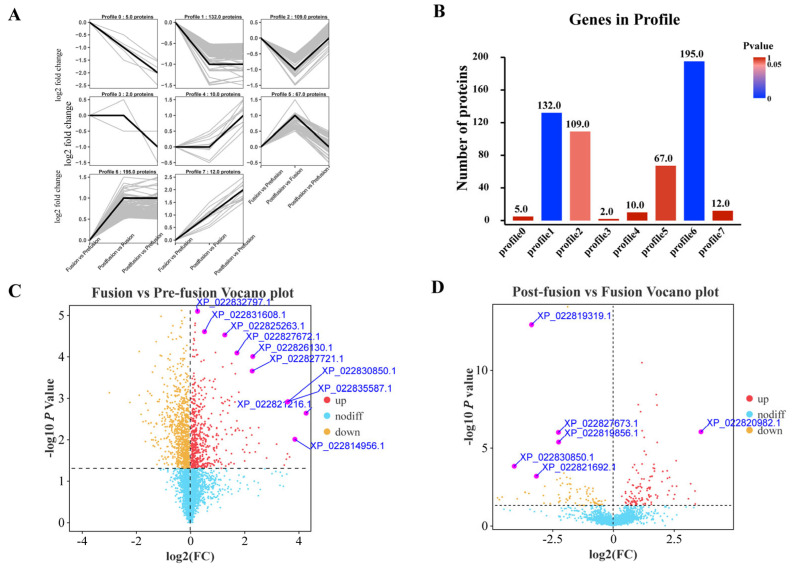
Proteomic sequencing data of the peritoneal sheath of *S. litura.* (**A**) Categorization of gene expression trends at pre-fusion, fusion, and post-fusion period. (**B**) Number of proteins of predominant protein expression patterns. (**C**) Volcano plot of differentially expressed proteins (DEP) during fusion versus pre-fusion period. (**D**) Volcano plot of DEP during post-fusion versus fusion period.

**Figure 5 ijms-26-05564-f005:**
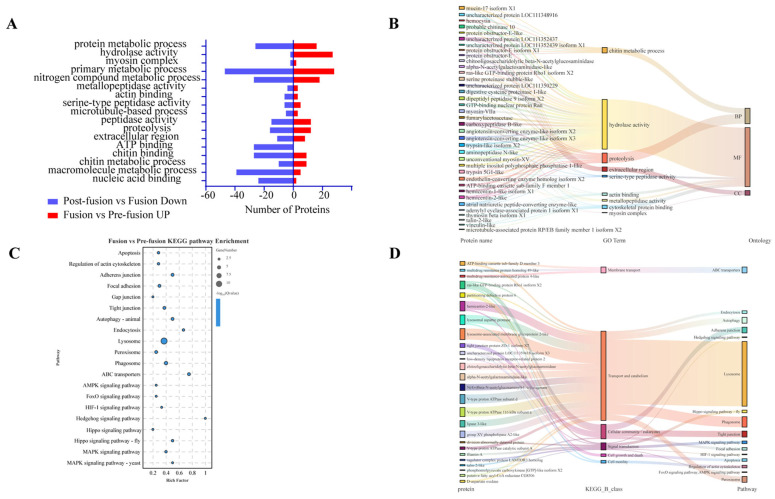
(**A**) Intersection of GO enrichment pathways for proteins in upregulated proteins during fusion compared to pre-fusion and downregulated proteins during post-fusion compared to fusion. (**B**) The Sankey diagram shows significant GO enrichment pathways and corresponding proteins during testicular fusion. (**C**) KEGG analysis of proteins upregulated during fusion compared to pre-fusion. (**D**) The Sankey diagram shows significant KEGG enrichment pathways and corresponding proteins during testicular fusion.

**Figure 6 ijms-26-05564-f006:**
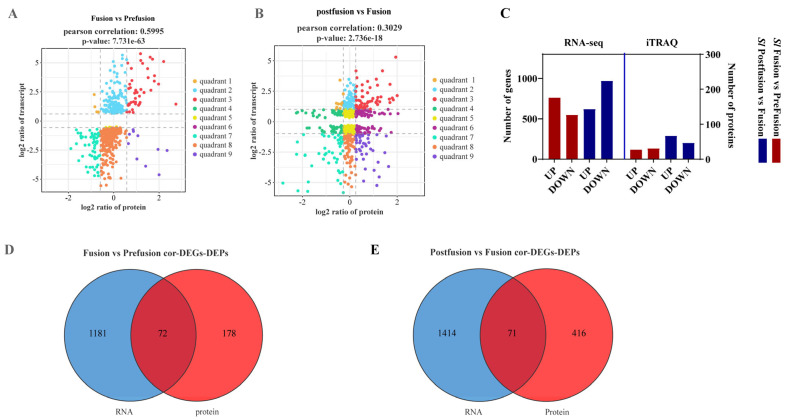
Nine-quadrant plot of proteome and transcriptome joint analysis. (**A**) Comparison of protein and mRNA expression levels in pre-fusion vs. fusion. (**B**) Comparison of protein and mRNA expression levels in post-fusion and fusion. Each point represents a gene/protein, with the Y-axis representing the log2 fold change of DEGs and the X-axis representing the log2 fold change of DEPs. Genes with expression changes of at least 2-fold (log2 = 1 or −1) are considered significantly differentially expressed, and proteins with expression changes of at least 1.5-fold are considered significantly differentially expressed. (**C**) Histogram illustrating significantly changed genes and proteins during the whole fusion process. (**D**,**E**) Venn diagrams of significantly changed genes and proteins during the whole fusion process.

**Figure 7 ijms-26-05564-f007:**
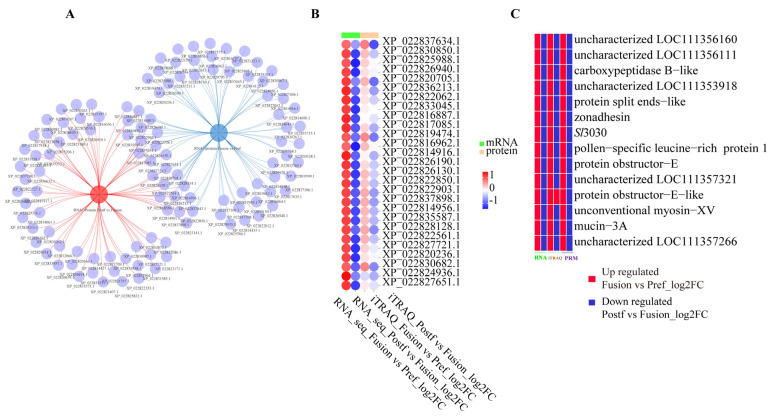
Screening of co-expressed genes and proteins. (**A**) Intersection of co-expressed genes and proteins between pre-fusion/fusion with post-fusion/fusion. The red connected area indicates genes and proteins commonly down-regulated after testicular fusion, and the blue connected area indicates genes and proteins commonly up-regulated during testicular fusion. (**B**) 27 genes/proteins that are highly expressed during the fusion stage at both the protein and mRNA levels. (**C**) 14 genes/proteins among the 27 that have been validated by parallel reaction monitoring (PRM).

**Figure 8 ijms-26-05564-f008:**
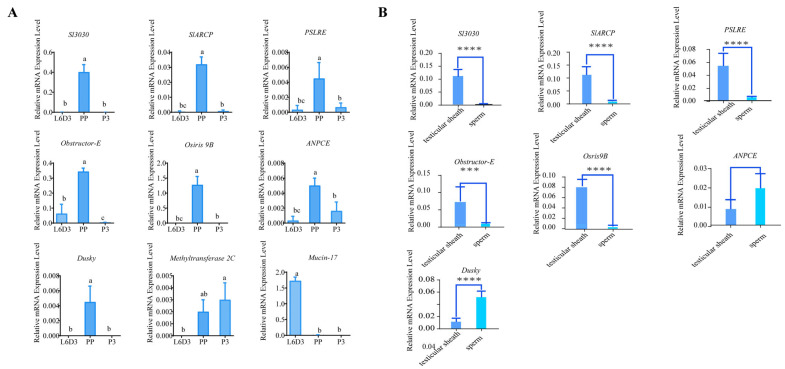
Quantitative PCR validation of gene expression and gene localization. (**A**) Verification of the temporal expression profile of the genes during testis development. Groups a,b,c sharing a common letter are not significantly different, while groups with no common letters are significantly different. (**B**) Verification of the localization of gene expression in the testicular sheath and sperm cells. *** indicates *p* value < 0.001, **** indicates *p* value < 0.0001.

**Figure 9 ijms-26-05564-f009:**
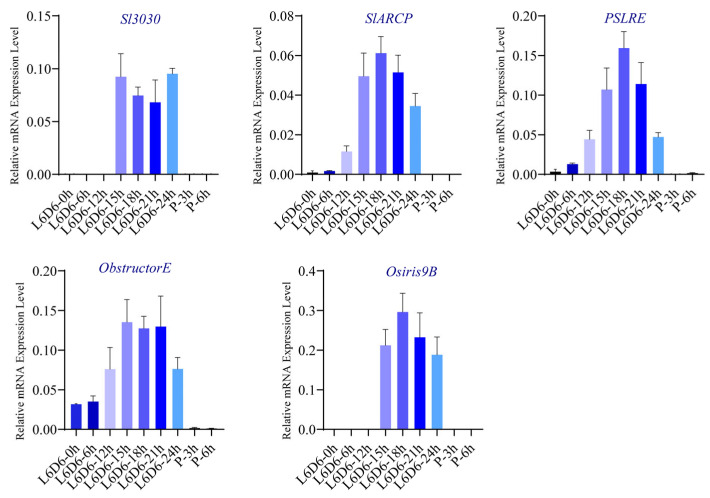
Gene expression during the critical testes fusion period within 24 h in *S. litura*.

**Figure 10 ijms-26-05564-f010:**
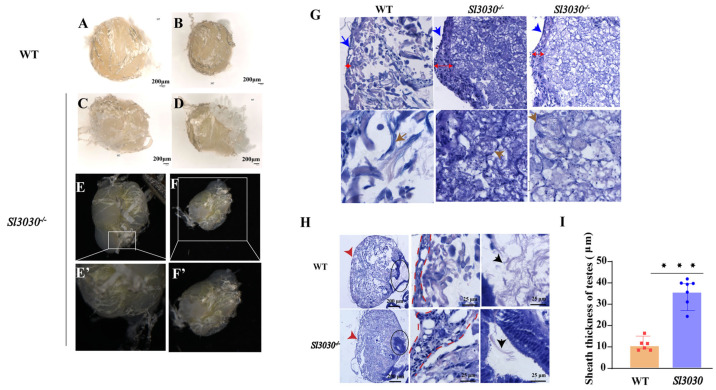
The impact of *Sl3030* gene deletion on testis fusion. (**A**,**B**) Wild-type testes; (**C**–**F**) Mutant testes. (**E**′,**F**′) are magnified views of (**E**,**F**), respectively. (**G**) Testicular sections that have not fused properly. The blue arrows indicate the position of the testicular peritoneal sheath; the gray arrows mark the sperm bundles and spermatids; the length of the red double-headed line segment represents the thickness of the peritoneal sheath. (**H**,**I**) Thickness comparison of the peritoneal sheath between wild-type and mutant testes. The red arrows indicate the location of the peritoneal sheath; the red dashed outline denotes the thickness of the sheath; the area within the black ellipse represents the vas deferens; the black arrows point to spermatids within the vas deferens. *** indicates *p* value < 0.001.

**Figure 11 ijms-26-05564-f011:**
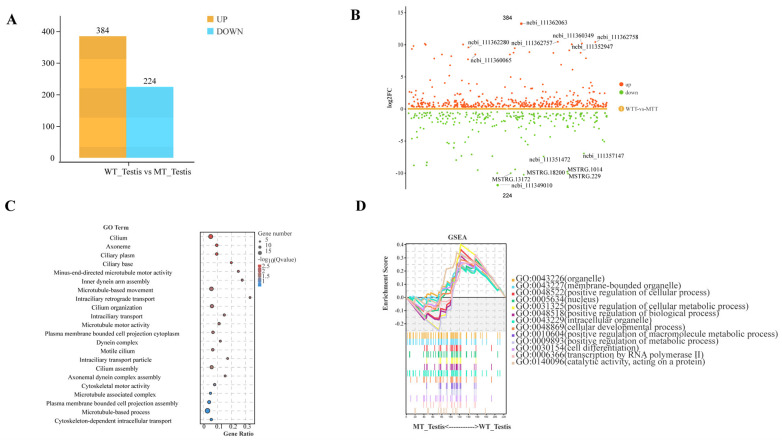
Transcriptome sequencing data of *Sl3030* mutant. (**A**) Bar chart of differentially expressed genes between wild-type and mutant. (**B**) Grouped difference scatter plot of differentially expressed genes between wild-type and mutant. (**C**) GO enrichment analysis of downregulated genes in the mutant type. (**D**) Gene Set Enrichment Analysis (GSEA) of downregulated genes in the mutant.

**Table 1 ijms-26-05564-t001:** List of proteins that are highly expressed at both the mRNA and protein levels during the testis fusion stage.

Accession	Gene_Id	Gene_Description	RNA-Seq_*Sl*_PP vs. L6D3_log2FC	Padj-1	RNA-Seq *Sl*_P3 vs. PP_log2FC	Padj-2	iTRAQ_log2FC	Padj-3
XP_022814918.1	111348513	ATP-dependent RNA helicase DHH1	10.83	4.29 × 10^−25^	−8.95	7.81 × 10^−31^	12.73	2.64 × 10^−2^
XP_022837634.1	111364820	serine/arginine repetitive matrix protein	3.98	2.73 × 10^−16^	−1.74	1.17 × 10^−3^	12.05	2.24 × 10^−2^
XP_022830850.1	111359506	acidic repeat-containing protein-like	10.80	1.50 × 10^−29^	−12.61	4.35 × 10^−32^	12.05	1.26 × 10^−3^
XP_022825988.1	111356021	uncharacterized LOC111356021	6.56	9.20 × 10^−10^	−7.78	3.29 × 10^−14^	5.29	2.02 × 10^−2^
XP_022826940.1	111356706	histone acetyltransferase p300-like	9.93	9.10 × 10^−17^	−11.16	1.53 × 10^−20^	3.84	1.77 × 10^−2^
XP_022820705.1	111352437	uncharacterized LOC111352437	3.87	1.37 × 10^−19^	−1.70	1.80 × 10^−4^	3.51	6.01 × 10^−3^
XP_022836213.1	111363605	atrial natriuretic peptide-converting enzyme-like	5.29	4.40 × 10^−8^	−2.20	2.11 × 10^−2^	3.30	3.60 × 10^−3^
XP 022822062.1	111353322	apolipoprotein D-like	5.47	9.86 × 10^−9^	−3.41	7.64 × 10^−5^	3.07	3.87 × 10^−2^
XP_022833045.1	111360972	glycine-rich cell wall structural protein-like	11.90	0.00	−12.61	1.40 × 10^−44^	2.84	3.61 × 10^−2^
XP_022816887.1	111349849	alpha-N-acetylgalactosaminidase-like	7.28	2.84 × 10^−9^	−3.92	7.47 × 10^−5^	2.60	1.10 × 10^−3^
XP_022817085.1	111349954	serine proteinase stubble-like	4.93	9.38 × 10^−6^	−3.65	6.49 × 10^−4^	2.44	1.22 × 10^−3^
XP_022819474.1	111351669	uncharacterized LOC111351669	1.57	4.84 × 10^−3^	−1.19	3.28 × 10^−2^	2.26	9.99 × 10^−3^
XP_022816962.1	111349883	NAD kinase 2C mitochondrial	2.12	5.43 × 10^−7^	−2.41	1.42 × 10^−10^	2.21	9.72 × 10^−4^
XP_022814916.1	111348511	polypeptide N-acetylgalactosaminyltransferase 2-like	3.46	5.18 × 10^−19^	−1.50	4.17 × 10^−12^	1.76	5.58 × 10^−4^
XP_022825632.1	111355800	uncharacterized LOC111355800	0.75	2.47 × 10^−2^	−0.72	1.69 × 10^−3^	1.50	1.55 × 10^−2^
XP_022826190.1	111356160	uncharacterized LOC111356160	7.70	1.39 × 10^−9^	−7.65	1.23 × 10^−9^	2.85	5.18 × 10^−3^
XP_022826130.1	111356111	uncharacterized LOC111356111	7.69	2.26 × 10^−6^	−10.88	1.83 × 10^−5^	4.96	8.52 × 10^−5^
XP_022822850.1	111353879	carboxypeptidase B-like	8.69	1.08 × 10^−10^	−7.47	9.86 × 10^−10^	9.12	1.30 × 10^−2^
XP_022822903.1	111353918	uncharacterized LOC111353918	7.53	2.37 × 10^−9^	−6.63	4.88 × 10^−8^	11.70	3.00 × 10^−2^
XP_022837898.1	111365016	protein split ends-like	6.01	6.48 × 10^−14^	−5.07	2.63 × 10^−11^	7.06	2.80 × 10^−2^
XP_022814956.1	111348538	zonadhesin	9.71	5.20 × 10^−19^	−9.45	1.23 × 10^−22^	14.57	9.95 × 10^−3^
XP_022835587.1	111363030	Sl3030	10.48	5.54 × 10^−16^	−10.74	3.34 × 10^−21^	12.21	1.22 × 10^−3^
XP_022828128.1	111357598	pollen-specific leucine-rich repeat extensin-like protein 1	7.27	0.00	−6.09	0.00	11.10	4.18 × 10^−2^
XP_022822561.1	111353686	protein obstructor-E	5.09	0.00	−5.75	0.00	4.64	5.29 × 10^−3^
XP_022827721.1	111357321	uncharacterized LOC111357321	11.34	2.14 × 10^−18^	−15.27	2.47 × 10^−17^	4.89	2.25 × 10^−4^
XP_022820236.1	111352109	protein obstructor-E-like	2.84	9.61 × 10^−25^	−2.49	7.28 × 10^−25^	1.68	4.39 × 10^−2^
XP_022830682.1	111359386	unconventional myosin-XV	1.43	1.96 × 10^−3^	−1.03	1.03 × 10^−2^	2.21	4.98 × 10^−2^
XP_022824936.1	111355348	mucin-3A	8.16	0.00E	−1.40	8.98 × 10^−3^	1.97	5.42 × 10^−3^
XP_022827651.1	111357266	uncharacterized LOC111357266	14.48	2.92 × 10^−10^	−14.12	2.07 × 10^−18^	4.02	2.04 × 10^−2^

## Data Availability

Data is contained within the article and Appendix A.

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
