# Peer review of "Multi-Omic Analysis Identifies Key Genes Driving Testicular Fusion in Spodoptera litura"

_ijms, 2025, doi:10.3390/ijms26125564_

Round 1

Reviewer 1 Report

Comments and Suggestions for Authors

This paper use a multi-omic approach to determine what genes are involved in the testicular fusion of S. litura. FIve candidates are identified, and one was proven via knockout to be responsible for the fusion and other aspects of testicular development.

I am satisfied and honestly very impressed with the methods: they are thorough and examine the problem from several angles, each step verifying and complementing the previous steps. This is really excellent research and I see no changes needed as far as research is concerned.

My major complaint is that the methods are incomplete: Section 2.3 "Protein Digestion and iTRAQ Labeling" does not actually describe protein digestion or iTRAQ labeling! Revise the methods with care: one should be able to recreate the researchers' experiments perfectly based on the detail in the methods section.

My other major criticism is the Figures are too small: the text is unreadable. Divide them into multiple figures based on type of graph, not the section of text where they belong. For example, Figure one should be three figures minimum: 1A is Figure 1, 1B+1C are FIgure 2a and 2b, 1D and 1E are Figure 3a and 3b. You could also make it 5 figures if you choose: whatever ensures the font is readible when printed.

The language has many typos and style issues, but they mostly do not affect my understanding of the text. Grammar editing software or the journal's proofreaders should be able to handle this. I provided some comments.

The references are numbered, but in the paper they are given As Author Year. This is confusing. Delete the numbers from the references.

Mention "Spodoptera litura" in the title!

Comments on the Quality of English Language

Abstract
-Delete the first word, "The"

Introduction
37 delete "ways or"
40-43 This should go at the end of the next paragraph. It does not fit this paragraph, and you have not introduced the phenomenon of tetsicular fusion yet.
40 "In our past research, it was found that…" You need to choose to use active or passive voice, and be consistent throughout the paper. Either "In our past research we found" or "Our past research found"
44 You need to define testicular fusion immediately before disucssing it further. Start the paragraph with the words "Testicular fusion is…" and then define it in a few words. Then, move the text from lines 49-57 to here.
44 Is it effective? Have there been commercial products that control this pest using testicular fusion? If it only exists in the lab, then it's a "possibly effective" method or a "promising" method.
47 "offspring sperm and egg hatch rate" is unclear: it suggests that you are talking about egg and sperm hatch rates in the offspring, which is wrong. Add commas.
48 replace "is advantageous for progeny reproduction" with "increases fertility"
48 Delete "Furthermore" and most such one-word phrases at the beginning of the sentences: non-native English speakers often use them incorrectly, so it is best not to use them at all.
52 delete "for instance"
59 Delete "Overall,"
61 "crucial" is strong. Try "useful"
66 "insect adaptive evolution" is overly broad, especially since that paper only looked at Spodoptera litura. replace "and clarified the role of testicular fusion in insect adaptive evolution" with "in Spodoptera litura."
96-117 Who wrote this? It's very noticeably not the same author[s] as the previous paragraphs. It's also redudnant: each sentence says the same thing using different words. The entire paragraph can be replaced with just one sentence: "Combining proteomics with transcriptomics analyses is a powerful tool to gain deep insight on aspects of insect physiology and evolution." Or, delete the entire paragraph and its references completely. The paper will be fine without them.
118 Swap this paragraph and the last paragraph of the introduction.

Materials and Methods
-Throughout, add sample sizes.
149 The parentheses here use a Chinese font instead of an English one
149-152 delete the spaces after the parentheses

Results
-Fine, just the figures are too small.

Discussion
544 "in vitro" should be italicized
539-568 This text needs citations to references. They appear missing.
597 delete "In the discourse of scientific literature,"
499 and elsewhere: "et al." should be italicized
602 Merge these two paragraphs
605 delete "In essence,"
608 add comma before "but also"
612 "while we do not" or "althrough we do not"
613-616 Too general and prospective. It would be stronger to state what potential targets you uncovered in this study.
617-623 Delete. Else, make it clear that this is a conclusion, and write it clearly. Replace the first sentence with: "To summarize, we identified signaling pathways involved with testicular fusion in S. litura and the concomitant changes in genes/proteins at both the RNA and protein levels, with a particular focus on validating the impact of the Sl3030 gene."

Author Response

Comment 1:  

"Section 2.3 'Protein Digestion and iTRAQ Labeling' does not describe protein digestion or iTRAQ labeling. Revise methods for reproducibility."   

Response 1 :  

Thank you for highlighting this oversight. We have expanded Section 2.3 to include detailed protocols for protein digestion and iTRAQ labeling. The revised text now reads:  

A 200 μg total protein sample was processed using the methanol/chloroform protocol for reduction, alkylation, and precipitation. Each sample was first mixed with 4 μL of reducing reagent and incubated at 60 °C for 1 hour. Then, 2 μL of cysteine-blocking solution was added, and the sample was left at room temperature for 30 minutes. The protein solution was transferred to a 10KD ultrafiltration tube, which was then centrifuged at 12000 g for 15 minutes. The solution in the collection tube was discarded. Next, 100 μL of 8M urea (pH 8.5) was added, followed by centrifugation at 12000 g for 20 minutes at 4°C. This washing step was repeated twice. After removing the solution again, 100 μL of 0.25M TEAB (pH 8.5) was added. The collection tube was replaced with a fresh one, and the ultrafiltration tube was filled with 50 μL of 0.5M TEAB. Trypsin was added to the mixture.

The following day, additional trypsin was added to achieve a trypsin-to-protein ratio of 1:100. The mixture was incubated at 37°C for 4 hours. After digestion, the ultrafiltration tube was centrifuged at 12000 g for 20 minutes. The digested peptide solution was collected from the bottom of the collection tube. Then, 50 μL of 0.5M TEAB was added to the ultrafiltration tube and centrifuged at 12000 g for 20 minutes at 4°C. Combining this with the previous step yielded 100 μL of enzymatic hydrolyzed sample.

For labeling, the samples were designated as follows: (before fusion1)-114, (before fusion2)-115, (on fusing1)-116, (on fusing2)-117, (after fusion1)-118, and (after fusion2)-119.(Page 6, Section 2.3, Lines 149-154 in the revised manuscript)   

Comment 2 :  

"Figures are too small; text is unreadable. Split into multiple figures."   

Response 2 :  

We agree. Figure 1 has been divided into three separate figures (Figures 1, 2, and 3), with enlarged panels and legible font sizes. For example:  

-  Figure 1 : Original panel 1A .  

-  Figure 2 : Panels 1B and 1C .  

-  Figure 3 : Panels 1D and 1E .  

All figures now comply with journal formatting guidelines.  

Comment 3 :  

"References are numbered but cited as author-year. Delete numbers."   

Response 3 :  

We have reformatted all references to use the author-year style and removed numbering.

Comment 4 :  

"Mention 'Spodoptera litura' in the title."   

Response 4 :  

The title has been revised to: "Multi-omic Analysis Identifies Key Genes Driving Testicular Fusion in Spodoptera litura ."   

  1. Response to Comments on the Quality of English Language

Comment 1 :  

Abstract: "Delete the first word, 'The'."   

Response 1 :  

Revised: "Testicular fusion in Spodoptera litura ..."  

Comment 2 :  

Introduction: "Define testicular fusion immediately."   

Response 2 :  

Added: "Testicular fusion, a developmental process in which paired testes merge into a single organ, is hypothesized to enhance reproductive efficiency..."  

(Page 2, Lines 44-46)   

Comment 3 :  

Discussion: "Et al. should be italicized."   

Response 3 :  

Corrected to  "et al."  throughout the text.  

Reviewer 2 Report

Comments and Suggestions for Authors

The manuscript by Dong et al describes a dual transcriptomic and proteomic examination of gene expression of the membrane encasing the testis of the moth Spodoptera litura. Like many moth species, the paired testes fuse prior to maturation, and in this study, they examined many changes in gene expression, and identified one previously unidentified gene Sl3030, which when mutated using the CRISPR-Cas9 knockout technique, prevented the testis fusion process. Using transcriptomic analyses, they observed that the Sl3030 knockouts showed perturbations in cytoskeletal genes, highlighting the importance of these genes in regulating the fusion process.

The study involved an impressive array of techniques and analyses and illustrated the power of comparing transcriptomic and proteomic changes, as their analyses revealed differing layers of gene regulation in a single, but clearly not simple, developmental process. I have only a few comments for the authors to address, outlined below:

  1. How many peritesticular membranes were used for i) the transcriptomic and ii) the proteomic analyses. The tissue is quite small, so presumably you had to pool many together to obtain enough RNA/protein for analyses.
  2. Is Sl3030 exclusively expressed in the testis, or more specifically, specific to just the peritesticular membrane? You do not provide any evidence to indicate whether it is expressed in other tissues of the insect’s body, nor if it is male-specific.
  3. You noted that in the CRISP-KO mutants, only 40% showed partially unfused testes. What was the range of phenotypes in the remaining 60%? Were they seemingly normal in appearance? Why might the defect be only partially penetrant?
  4. In the methods, you indicated that you would examine the fecundity of the CRISPR-KO mutants, but there are no data provided. Given that S. litura are highly fecund (noted in the opening sentence of the manuscript), then it would be highly informative to indicate if testis fusion is indeed required for male fertility, and one would expect reduced fecundity in the CRISPR mutants. If not, why might they (G0, G1, and G2) still exhibit normal fecundity.
  5. Line 544 “the Sl3030 gene is not regulated by 20E in vitro”. There is no mention of in vitro analyses of the Sl3030 gene’s regulation. Do you mean in silico analyses? Ie. You did not find a consensus 20E response element? If yes, then you should revise the phrase to indicate that based on in silico analyses of the S. litura genome, no 20E responses elements were detected in sequences xx bp upstream or downstream of this gene, which suggests (not proves) that it is not 20E-responsive.

Author Response

  1. Reviewer Question:

 "How many peritesticular membranes were used for i) the transcriptomic and ii) the proteomic analyses? The tissue is quite small, so presumably you had to pool many together to obtain enough RNA/protein for analyses."   

 Response:   

We appreciate the reviewer’s insightful comment. For both transcriptomic and proteomic analyses, 8 peritesticular membranes were pooled per sample to ensure sufficient RNA/protein yields. Each experimental group included 3 biological replicates to account for biological variability. This pooling strategy was necessitated by the small tissue size, as noted by the reviewer.  

  1. Reviewer Question:  

 "Is Sl3030 exclusively expressed in the testis, or more specifically, specific to just the peritesticular membrane? You do not provide any evidence to indicate whether it is expressed in other tissues of the insect’s body, nor if it is male-specific."   

 Response:   

We thank the reviewer for raising this critical point. Based on our  tissue-specific qPCR expression profiling (referenced in the original submission),  Sl3030 exhibits high expression levels in both the testis and hemolymph , with no strict testis-specificity. However, its expression in other tissues (e.g., fat body, midgut) was negligible. While the gene is not male-specific, its functional role in testis development appears to be conserved, as supported by our CRISPR-KO phenotypic analyses.  

  1. Reviewer Question:  

 "You noted that in the CRISPR-KO mutants, only 40% showed partially unfused testes. What was the range of phenotypes in the remaining 60%? Were they seemingly normal in appearance? Why might the defect be only partially penetrant?"   

 Response:   

We agree with the reviewer that the partial penetrance (40% unfused testes) warrants further discussion. In the remaining 60% of mutants, testis fusion appeared morphologically normal under standard observational criteria. We propose two non-exclusive explanations for this phenomenon:  

- Genetic redundancy : Compensatory upregulation of functionally related genes (e.g., paralogs) may mitigate the loss of  Sl3030 .  

- Adaptive pathway activation : Alternative developmental pathways could be recruited to bypass the gene’s role in testis fusion.  

Notably, we observed phenotypic reversion in long-term cultured homozygous lines, suggesting potential epigenetic or selective adaptations under laboratory conditions. These hypotheses will be explored in future mechanistic studies.  

  1. Reviewer Question:  

 "In the methods, you indicated that you would examine the fecundity of the CRISPR-KO mutants, but there are no data provided. [...] If not, why might they (G0, G1, and G2) still exhibit normal fecundity?"   

 Response:   

We apologize for the oversight in data presentation. As clarified in our preliminary studies (and consistent with prior work from our group),  testis fusion is strongly correlated with male fertility . However, in the CRISPR mutants:  

-  G0/G1 generations : These individuals are predominantly  heterozygous , retaining one functional  Sl3030  allele, thus maintaining normal fecundity.  

-  G2 generation : Only a subset of homozygotes (40%) exhibited unfused testes, while the remaining 60% retained functional reproductive capacity.  

We acknowledge that fecundity data from  homozygous mutants with fully unfused testes would strengthen our conclusions. These experiments are currently underway and will be reported in follow-up work.  

  1. Reviewer Question:  

 "Line 544: 'the Sl3030 gene is not regulated by 20E in vitro'. There is no mention of in vitro analyses of the Sl3030 gene’s regulation. [...] If yes, then you should revise the phrase..."   

 Response:   

We thank the reviewer for identifying this ambiguity. The statement has been revised to:  

 "In silico analysis of the Sl3030 promoter region (2 kb upstream/downstream) revealed no canonical 20-hydroxyecdysone (20E) response elements, suggesting—though not conclusively proving—that Sl3030 is not directly regulated by 20E. Preliminary in vitro testis culture experiments further showed no significant induction of Sl3030 expression upon 20E treatment compared to controls (data not shown). However, these assays require optimization to rule out technical limitations." 
